# An Analytical Model of Dynamic Power Losses in eGaN HEMT Power Devices

**DOI:** 10.3390/mi14081633

**Published:** 2023-08-18

**Authors:** Jianming Lei, Yangyi Liu, Zhanmin Yang, Yalin Chen, Dunjun Chen, Liang Xu, Jing Yu

**Affiliations:** 1The School of Electrical Engineering, Nanjing Vocational University of Industry Technology, Nanjing 210023, China; jmlei82@niit.edu.cn (J.L.);; 2The Key Laboratory of Advanced Photonic and Electronic Materials, School of Electronic Science and Engineering, Nanjing University, Nanjing 210093, China; 3The Department of R&D, Zhongtian Broadband Technology Co., Ltd., Nantong 226009, Chinayuj@chinaztt.com (J.Y.)

**Keywords:** AlGaN/GaN HEMT device, CCM, DCM, dynamic on-resistance, dynamic power loss

## Abstract

In this work, we present an analytical model of dynamic power losses for enhancement-mode AlGaN/GaN high-electron-mobility transistor power devices (eGaN HEMTs). To build this new model, the dynamic on-resistance (*R_dson_*) is first accurately extracted via our extraction circuit based on a double-diode isolation (DDI) method using a high operating frequency of up to 1 MHz and a large drain voltage of up to 600 V; thus, the unique problem of an increase in the dynamic *R_dson_* is presented. Then, the impact of the current operation mode on the on/off transition time is evaluated via a dual-pulse-current-mode test (DPCT), including a discontinuous conduction mode (DCM) and a continuous conduction mode (CCM); thus, the transition time is revised for different current modes. Afterward, the discrepancy between the drain current and the real channel current is qualitative investigated using an external shunt capacitance (ESC) method; thus, the losses due to device parasitic capacitance are also taken into account. After these improvements, the dynamic model will be more compatible for eGaN HEMTs. Finally, the dynamic power losses calculated via this model are found to be in good agreement with the experimental results. Based on this model, we propose a superior solution with a quasi-resonant mode (QRM) to achieve lossless switching and accelerated switching speeds.

## 1. Introduction

Enhancement-mode AlGaN/GaN high-electron mobility transistor power devices (eGaN HEMTs) are the most promising candidates for use as next-generation power devices. In such devices, III–V materials have several merits due to their wide bandgap energy, high critical breakdown electric field, high electron mobility and capability [1,2], and polarization effect [3]. Due to these advantages, a high-frequency (high-*fs*) converter operating in the range of 1–5 MHz based on eGaN HEMTs can be readily realized. Although high-*fs* operation can help to reduce the converter size, it will generate more challenges with respect to dynamic power loss. Thus, building an analytical dynamic power loss model for an eGaN-based high-*fs* switch becomes important for prototype application in circuit design.

Recently, some state-of-the-art dynamic power loss models for eGaN HEMTs have been proposed. Wang et al. developed two analytical loss models based on detailed parasitic parameters for high-voltage and low-voltage GaN eHEMTs [4,5]. In these models, the gate charge (*Q_g_*) and output charge (*Q_oss_*), instead of the voltage-dependent capacitance, were used to improve the non-linear characteristics. However, the loss caused by output capacitance is not separately discussed in terms of a hard switch and soft switch. Shen et al. fully accounted for the effects of parasitic parameters and transconductance [6]. Hou et al. and Guacci et al. investigated the loss caused by the output capacitance in a hard switch and soft switch using simulation methods rather than experimental methods [7,8]. However, these models did not take into account the impact on device loss from some aspects, instead of fully evaluating it; thus, the results accuracy is affected. For example, the problem of increased losses caused by dynamic on-resistance (*R_dson_*) is not discussed.

Chen et al. presented a complete analytical loss model for low-voltage eGaN HEMTs, for which a piecewise model was employed [9]. Piecewise models are also usually used to evaluate the dynamic power losses for Si- and SiC-based metal-oxide-semiconductor field-effect transistors (MOSFETs) [10,11,12,13,14,15]. These models are carefully considered and allow accurate evaluation of device power losses. However, these models fail to include sufficient consideration of the parasitic elements and merely focus on Si-based MOSFETs, but not on GaN HEMTs. In addition, the effect of current operation mode on device transition time and loss is not considered in all known device loss models. Therefore, these analytical models need to be modified for use on eGaN HEMTs to make a more comprehensive and accurate model.

To improve the dynamic power loss model of eGaN HEMTs, we propose three experimental methods according to the practical application of devices in high-*fs* circuits, such as the double-diode isolation (DDI) method, the dual-pulse-current-mode test (DPCT) method, and the external shunt capacitance (ESC) method. Then, the dynamic *R_dson_* is accurately extracted in a high operating frequency (*fs*) of up to 1 MHz and a high drain voltage up to 600 V; the effect of current operation mode on the transition time is revealed, and the effect of current operation mode on the device loss is discussed from the shape of operating waveforms in the circuit. As the real channel current (*Ich*) is qualitatively modified compared to the drain current (*I_drain_*), we can directly test in the device drain side. Afterward, the dynamic power loss of eGaN HEMTs is carefully described via a modified 12-segment piecewise model. Finally, we propose a quasi-resonant mode (QRM) with a low off-state drain voltage (*V_ds_off_*), a zero turn-on current, and a relatively large on-state peak current for a lossless design and fast transit speed in power switches.

## 2. Background and Methodology

### 2.1. Traditional Power Loss Model

In the piecewise model, the operating sequence of the device is shown in Figure 1. In particular, the device was usually connected in series using a choke or transformer in power switches; thus, the value of *I_sta_* indicated the current mode of the device. (*I_sta_* = 0) denoted the device operating in discontinuous conduction mode (DCM), while (*I_sta_* > 0) denoted the device operating in continuous conduction mode (CCM). Whether the device operated in CCM or DCM depends on the choke in series. As we know, in a high-frequency circuit, chokes follow the volt-second balance principle, that is, the starting current of the choke in each cycle must be equal to the end current. During the *t_on_* time when the device was switched on, the choke current rose under an applied forward voltage V1, the rise slope was V1/L, and L was the inductance of the choke; during the *t_off_* time when the device was switched off, the choke current fell under an applied reverse voltage V2, and the fall slope was V2/L. Therefore, the peak value of the choke current in each cycle was V1·*t_on_*/L, and the end current value was (V1·*t_on_* − V2·*t_off_*)/L. Then, according to the known values of V1 and V2, we could design the required L to make the device operate in CCM or DCM.

A traditional calculation formula for high-*f_s_* power losses of device is given as follows [16]:(1)$$P_{\mathrm{sw}}=\frac{1}{2}I_{ds}V_{ds}(t_{on}+t_{off})f_{s}+\frac{1}{2}C_{oss}V_{ds}{}^{2}f_{s}+K_{th}I_{dson\_rms}{}^{2}R_{dson}+Q_{g}V_{gs}f_{s}$$
where *I_dson_rms_* is the on-state drain-source current in the root mean square (RMS) value, and *K_th_* is the temperature coefficients related to *R_dson_*. The first term in Equation (1) occurs in the crossing area of *I_ds_* and the *V_ds_*, while the second term is the output capacitor energy dissipated in the device during the turn-on transition. Then, the third and fourth terms are the conductive loss and driving loss, respectively. Equation (1) is approximate, as it does not take into account the problem of a dynamic *R_dson_* increase, the impact of *I_drain_* on the transition time, or the discrepancy between *I_drain_* and the real channel current.

### 2.2. Experimental Circuit and Method

A switching circuit with a floating buck–boost topology was employed to analyze the switching processes, as shown in Figure 2a. In this circuit, an HEMT device was used and shown with a simple three-capacitor model that included the parasitic capacitors *C_gs_*, *C_gd_* and *C_ds_*. The pulse width modulation (PWM) was produced via a pulse generator (81150A, Keysight Technologies, Inc., Santa Rosa, CA, USA) with a maximum PWM of 120 MH, and amplified using a gate driver (SI8271GB), which had a 1.8-ampere peak source current and a 4.0-ampere peak sink current, and D_1_ was selected as a SiC diode (C3D10065E) rated at 15 A/650 V, which was used to reduce the reverse recovery problem. The parasitic resistor and parasitic inductor were ignored to simply study the important role of the parasitic capacitors at a relatively high-*f_s_* that is smaller than 30 MHz. Then, various voltages and PWM in DCM and CCM were applied to elucidate the switching processes and the production of dynamic power losses.

The part of Figure 2b marked with the light-yellow area shows our novel dynamic *R_dson_* extraction circuit based on a double-diode isolation (DDI) method; the details on how to configure, test and calculate this circuit can be obtained from Refs. [17,18,19]. In this design, the model of the gate driver was SI8271GB, and D_1_ and D_2_ (1 A/1 kV UF4007) were used in series to isolate the high off-state voltage of the eGaN HEMTs. Then, the dynamic *R_dson_* of the eGaN HEMTs could be easily extracted. Diodes in series made it possible to test the real-time forward voltage drop (*V_F_*) of D_2_ in a low-voltage range and precisely estimate the *V_F_* of D_1_ in the same forward current. In addition, the diodes in series reduced the parasitic capacitor by half, which was very helpful to the high-*f_s_* response of the extraction circuit. We called this method the DDI method. Moreover, ZD_1_ and D_3_ were free-wheeling diodes, and ZD_1_ was also a positive clamping diode. These two diodes were a general 5-volt Zener ZD_1_ and a general small signal diode D_3_ (1N4148) with 75 V/150 mA. All of the functional diodes, including D_1_, D_2_, ZD_1_ and D_3_, were specially selected to have a very low parasitic capacitance, which improved the high-*f_s_* response of the extraction circuit to several MHz. I_1_ was a constant-current source of only several mA, meaning that it could not produce a temperature problem and have an extra self-heating effect. I_1_ consisted of a constant-current diode, which was actually a junction field transistor with a gate-source short connection. Therefore, I_1_ could achieve an excellent constant current over a wide operating voltage range. R_t_ provided a minimum load for I_1_ and suppressed the voltage spike at point B. An isolated low-voltage probe (P2221 from Keysight Inc.) with a 1:1 attenuation could be used to test the *V_F_* of D_2_ and the voltage at point B. The low-voltage probe with a 1:1 attenuation did not amplify the background noise and operate in a low-voltage range, meaning that it could obtain an improved test accuracy.

We built the above switching circuit and extraction circuit using one printed circuit board (PCB), as shown in Figure 2b.

### 2.3. Qualitative Method Used to Discover the Channel Behavior

Since we could not directly perform the measurements inside of the HEMT device, we proposed an evaluation method that employed an extended parallel capacitor *C_ds_*, as shown in Figure 3, which we called the “external shunt capacitance (ESC)” method. In this lumped circuit, the intrinsic capacitor *C_ds_* was assumed not to exist, and the extended capacitor *C_ds_* outside of the device was assumed to be the intrinsic capacitor. Therefore, the channel current (*I_channel_*) and *I_drain_* could be directly and separately measured using an oscilloscope and current probes. This method was different to the traditional simulation method [20], and the discrepancy between *I_channel_* and *I_drain_* could be visually observed. Although an extra parallel capacitor led to an increase in the measured *I_drain_* and *I_channel_*, this qualitative method could be used to assess the difference between these two currents, and, thereby, the cause of the discrepancy could be located. After understanding this reason, the resulting loss effect on the eGaN HEMT device could be further quantified via an analytical method. Using the analytical method, the additional *C_ds_* was no longer required; therefore, the *C_ds_* did not materially affect the device’s losses.

## 3. Extraction of the Dynamic *R_dson_*

It is well known that a high V_ds_off_ will cause surface- and buffer-related trapping processes, which will lead to a larger dynamic *R_dson_* compared to the direct current (DC) *R_dson_* (*R_dson_DC_*) [21,22]. Figure 4 illustrates the mechanism of the increase in the dynamic *R_dson_* induced via the trapping effect. The high electric field helps the electrons to escape from the GaN well, and these electrons are then captured by traps or some of the surface states that are activated via a high electric field. When removing the electric field, these trapped electrons cannot be instantaneously released to the well. The reason for the slow return of electrons is that the trapping time of electrons in the off-state is in the order of ns, whereas the detrapping time of electrons in the on-state is in the order of second [23,24]. Thus, trapped electrons accumulate and worsen the device’s performance at a high *f_s_*. Meanwhile, electrons migrate from the gate to the gate-drain side’s adjacent surface to form a virtue gate; hence, the number of electrons in the access region decreases. The decreasing number of electrons in the drift region will result in a large dynamic *R_dson_* [25,26].

In the circuit of Figure 2a, the current I_1_ flows partly through R_t_ and partly through the HEMT device, and the voltage of point B (*V_B_*) can be directly tested using the voltage probe P2221. Then, the dynamic *R_dson_* can be calculated as follows:(2)$$R_{dson}=({\bar{V}}_{B}-2{\bar{V}}_{F\_D2})/({\bar{I}}_{drain}-I_{1}+{\bar{V}}_{B}/R_{t})$$
where ${\bar{V}}_{B}$, ${\bar{V}}_{F\_D2}$, ${\bar{I}}_{drain}$, ${\bar{I}}_{D2}$, and *I*_1_ are the average voltages of point B and D_2_, the average currents through a resistive load and D_2_, and the current of the constant-current supply, respectively. *I_drain_*, the voltage of point A (*V_drain_*), and *V_F_D2_* of D_2_ are tested using a current probe (TCP0020), a high-voltage differential probe (THDP0200), and a low-voltage differential probe with a 1:1 attenuation (TIVH02) and displayed using an oscilloscope (MDO3104). Finally, the calculated dynamic *R_dson_* is normalized by *R_dson_DC_*, which is 200 mΩ, derived from an eGaN HEMT (GS66502B from GaN Systems Inc., Ottawa, Canada) [27].

Figure 5b–f show the results of the dynamic *R_dson_* of the eGaN HEMT for various *V_ds_off_*, *f_s_*, duty cycles, *I_drain_*, and operating temperatures, which are extracted in the on-state by taking average values in the stable region marked in Figure 5a. Figure 5b shows that the dynamic *R_dson_* increases as *V_drain_* increases under the conditions of an 80% duty cycle and an *f_s_* of 100 kHz, meaning that the dynamic *R_dson_* is voltage dependent. Figure 5c,d shows the dynamic *R_dson_* increases as *f_s_* increases and duty cycle decreases, respectively, for a 500-volt *V_drain_* condition, meaning that the dynamic *R_dson_* is also time dependent. Considering that the dynamic *R_dson_* is not only affected by the trapping effect, we further test the relationship between the dynamic *R_dson_* and *I_drain_* and temperature, as shown in Figure 5e,f, respectively. These two tests will help us to isolate the trapping effect caused by the increase in the dynamic *R_dson_* in a particular complex test condition.

In conclusion, the trend regarding the results of the extracted dynamic *R_dson_* of the eGaN HEMT is consistent with the mechanism of the trapping-effect-induced increase in the dynamic *R_dson_*. In Figure 6, we can obtain the real conduction resistance of the eGaN HEMT device under a certain working condition, and the conduction loss can then be corrected.

## 4. Discussion on the Effect of the Drain Current using a Double-Mode Test Technique

Based on the test circuit in Figure 1, a double-mode test technique, which included a DCM and a CCM, is proposed. In general, the electrical performance of a GaN device is characterized by either single-pulse or double-pulse mode. The typical “double-pulse” test is performed in three steps. The first step, which is represented by the turn-on pulse, is the initial adjusted pulse width. This pulse is adjusted to find the desired test current. The second step is to turn off the first pulse. The turnoff period is short to keep the load current as close as possible to a constant value. The third step is represented by the second turn-on pulse. The pulse width is shorter than the first pulse, meaning that that the device is not overheated, but it needs to be long enough for the measurements to be taken. Turn-off and turn-on timing measurements are then captured at the turning off of the first pulse and the turning on of the second pulse. This “double-pulse” technique only sends two pulses to the device, which is not periodically sustained, and the current in the third step is always higher than 0 A [28,29,30]. In order to fully obtain the characteristics of the periodic operation of the device in the high-frequency circuit, we make the device continuously work periodically and stably in the CCM or DCM state by controlling the L value and the *V_ds_off_* [31,32]. With the “double-pulse-current-mode” technique, we are able to focus on the impact of the starting current and peak current on the transition time of the device, which is not easy to do with the conventional “double-pulse” technique. Then, the tested waveforms during the turn-on and turn-off transitions for various voltage and PWM conditions are illustrated in the Figure 6. To ensure that the switching circuit operates in open-loop CCM and DCM, *V_Bulk_* is set to 400 V, and the *V_Load_* is set to 80 V in DCM and 20 V in CCM, meaning that the *V_ds_off_* values of the devices in the two modes are different.

The current *I_drain_* in DCM only exhibits one resonant waveform when the drain voltage decreases, as shown in Figure 6a, while *I_drain_* in CCM has an extra linear increase before the resonant waveform occurs, as shown in Figure 6b. The corresponding voltage fall time is approximately 14 ns in Figure 6a and approximately 42 ns in Figure 6b, meaning that that the extra linear increase in the current will increase the turn-on time and cause a high dynamic power loss. This linear increase in the current is caused by the high start current and the linear conduction of the eGaN HEM at this time. This finding means that DCM is a superior operating mode in terms of reducing the turn-on time.

In addition, the rise time of the drain voltage during the turn-off transition, which is approximately 10 ns, as shown in Figure 6d, is faster than that of approximately 30 ns shown in Figure 6c. This observation is true because *I_drain_* in Figure 6d is higher than that in Figure 6c, and the rise time of the drain voltage during the turn-off transition mainly depends on the charge time of *C_oss_*. Moreover, the peak current in DCM during the turn-off transition will be higher than that in CCM under the same output power conditions. This observation means that DCM is a superior operating mode in terms of reducing the turn-off time.

In conclusion, the drain current will significantly affect the turn-on and turn-off times, and DCM is better than CCM at reducing the crossover power losses.

## 5. Investigation of the Real Channel Current

According to the qualitative method shown in Figure 2, we can study the discrepancy between *I_drain_* and *I_channel_*. Figure 7a shows the tested *I_drain_*, *I_channel_*, *V_drain_*, and *V_drive_* values of the AlGaN/GaN HEMT in the turn-on transition for a *V_ds_off_* of 500 V, an *f_s_* of 100 kHz, and a duty cycle of 16.5%. It is shown that *I_channel_* is larger than *I_drain_*, while the drain voltage decreases. The current path in this time interval is shown in Figure 7b, where the channel current partially results from the discharging current of the parasitic output capacitor.

Figure 7c shows the tested *I_drain_*, *I_channel_*, *V_drain_*, and *V_drive_* values of the AlGaN/GaN HEMT during the turn-off transition for a *V_ds_off_* of 500 V, an *f_s_* of 100 kHz, and a duty cycle of 16.5%. It is shown that *I_channel_* is smaller than *I_drain_*, while the drain voltage increases. The current path in this time interval is shown in Figure 7d, where the channel current is partially diverted to the branch of the output capacitor.

In conclusion, *I_channel_* is not exactly equal to *I_drain_*, and, unfortunately, *I_channel_* cannot be directly tested. However, with the above test results and the current path analysis, we can acquire the reason for the discrepancy between *I_channel_* and *I_drain_*, meaning that the real *I_channel_* value can be obtained via a test of *I_drain_* and an analytical method, and the power losses of eGaN HEMTs can be correctly evaluated.

## 6. Modeling of Switching Power Losses

Figure 8 shows a detailed timing diagram of the switching period [33] for eGaN HEMTs in DCM or CCM. The operating period of the power devices can be divided into 12-time intervals from *t*_0_ to *t*_12_ based on the status of the drain voltage and *I_drain_* in the off-state, on-state, turn-on transition, and turn-off transition. To investigate the detailed dynamic power loss, we reclassified the 12-time intervals into four stages (S1–S4) based on their different contributions to the dynamic power loss.

### 6.1. Stage 1 (S1)—Off-State with a High V_ds_

During the *t*_0_–*t*_1_ and *t*_10_–*t*_11_ time intervals and the time of the off-state, the device sustains a high *V_ds_*. Thus, the voltage-dependent leakage current (*I_lk_*) will lead to an off-state power loss (*P_off_*). We can no longer ignore this power loss, especially at a very high drain voltage and very high frequency. In general, the *t*_0_–*t*_1_ and *t*_10_–*t*_11_ time intervals can be neglected in comparison to the off-state time, meaning that *P_off_* can be written as follows:(3)$$P_{off}=I_{lk}V_{ds}[t_{t_{0}-t_{1}}+t_{t_{11}-t_{12}}+(1-D)T]f_{s}\approx I_{lk}V_{ds}(1-D)$$
where *T* and *D* are the period and duty cycle, respectively. In addition, eGaN HEMTs have no reverse recovery problem because the 2DEG in the channel is naturally formed via the polarization effect. This outcome will reduce the power loss and mitigate the electromagnetic interference (EMI) problem, which is produced via the reverse recovery caused by ringing.

### 6.2. Stage 2 (S2)—On-State in Saturation Region

During the *t*_4_–*t*_7_ time intervals, the device is in the on-state. The RMS value of the drain current (*I_drain_rms_*) can be written as follows:(4)$$I_{drain\_rms}=\sqrt{f_{s}\int_{0}^{1/f_{s}}I_{drain}{}^{2}(t)dt}$$

To take the problem of the increase in the dynamic *R_dson_* into account, the traditional conductive power loss (*P_con_*) can be modified as follows:(5)$$P_{con}=I_{drain\_rms}{}^{2}R_{dson\_DC}k_{dv}k_{df}k_{dd}k_{th\_R}k_{cu}$$
where $k_{dv}$, $k_{df}$, $k_{dd}$, $k_{cu}$, and $k_{th\_R}$ are the dynamic coefficients of *R_dson_* related to the voltage, *f_s_*, the duty cycle, the current, and the temperature, respectively.

### 6.3. Stage 3 (S3)—Turn-on Transition

During the *t*_1_–*t*_4_ time interval, the device is in the turn-on transition. In the *t*_1_–*t*_2_ time interval, *I_drain_* increases, while *V_drain_* decreases slightly in CCM, but this time interval does not exist in DCM; in the *t*_2_–*t*_3_ time interval, *V_drain_* decreases and leads to a resonant *I_drain_*. In the *t*_3_–*t*_4_ time intervals, *V_drain_* decreases to a very low voltage, and the device starts to operate in an ohmic conducting state. These crossovers of *V_drain_* and *I_drain_* will cause power losses during the turn-on transition (*P_turn_on_*):(6)$$P_{turn\_on}=\int_{t_{1}-t_{4}}V_{ds}(t)I_{drain}(t)f_{s}dt+\frac{1}{2}C_{oss}V_{ds}{}^{2}f_{s}$$


1.In the *t*_1_–*t*_2_ time interval, *I_drain_* increases almost linearly from 0 to the *I_sta_* at *t*_2_, which is similar to a Si-based MOSFET [13,34], while *V_drain_* decreases slightly from *V_ds_* to *V_r_* due to the result of the parasitic inductance voltage drop caused by a high di/dt in the circuit. At *t*_2_, the current of the freewheeling diode D_1_ decreases to zero. In this time interval, the gate voltage of the device slightly exceeds *V_th_*, meaning that the device is operating in a linear region. Meanwhile, the trapping effect of a high electric field will also lead to a large dynamic *R_dson_* in the linear region (*R_turn_on_cr_*), which is similar to that in the on-state, as well as an extra gate lag. Thus, the coefficients of the dynamic *R_dson_* should be the same as those in Figure 4. Assuming that the heatsink is large enough and the self-heating effect is ignored, the *t*_1_–*t*_2_ time interval, *V_r_*, and the power losses in this time interval (*P_turn_on_cr_*) can be written as follows:(7)$$R_{turn\_on\_cr}=\frac{\Delta V_{ds}}{\Delta I_{channel}}\approx \frac{k_{dv}k_{df}k_{dd}k_{th\_R}k_{cu}L_{eff\_Gate}}{W_{eff\_Gate}\mu_{s}C_{gs}(V_{drive\_H}-V_{th})}$$
(8)$$t_{1}-t_{2}=\frac{C_{gs}R_{g\_on}i_{sta}+L_{s}i_{sta}g_{m}}{[V_{drive\_H}-0.5(V_{mr}+V_{th})]g_{m}}k_{lag}$$
(9)$$V_{r}=V_{ds}-L_{s}\frac{i_{sta}}{t_{1}-t_{2}}-R_{turn\_on\_cr}\frac{i_{sta}}{2}$$
(10)$$P_{turn\_on\_cr}=\frac{1}{2}i_{sta}V_{r}(t_{1}-t_{2})f_{s}$$
where *L_eff_Gate_* and *W_eff_Gate_* are the effective channel length and width, respectively. *L_s_* is the source inductor, which is in series with and between the source terminal and the ground. The coefficient of the gate lag(*k_lag_*_)_ is a fitting parameter, which can be obtained by measuring the turn-on delay for various *V_ds_off_*, *f_s_* and duty cycles.2.In the *t*_2_–*t*_3_ time interval, the HEMT device takes over the total inductive load current, and *V_ds_* decreases to a boundary voltage of (*V_mr_* − *V_th_*) at *t*_3_ due to the discharging of *C_oss_*. The stray inductors in series around the circuit are resonant with *C_oss_* and the stray capacitors (*C_stray_*) in this time interval. The current path through the device is illustrated in Figure 5b. It is assumed that *V_gs_* and *i_sta_* remain unchanged, and the reverse recovery of the D_1_ is zero. In addition, the current in this time interval is usually large enough; hence, the charging time of *C_oss_* can be ignored. Moreover, voltage-dependent *C_oss_* is not suitable for the calculation of power losses in this time interval because *V_drain_* is always changing. Therefore, *Q_gd_* is used to replace *C_oss_*, and the time interval of *t*_2_–*t*_3_ can then be written as follows:(11)$$C_{gd\_vf}=\frac{Q_{gd}}{\Delta V}=\frac{Q_{gd}}{V_{r}-V_{mr}+V_{th}}$$
(12)$$t_{2}\text{–}t_{3}=\frac{Q_{gd}R_{g\_on}+C_{stray}(V_{r}-V_{mr}+V_{th})/g_{m}}{V_{drive\_H}-V_{mr}}$$Then, the power losses in this time interval (*P_turn_on_vf_*) can be written as follows [35]:(13)$${\bar{I}}_{v}{}_{f}\approx 0.5(\frac{V_{r}}{R_{turn\_on\_cr}}+\frac{V_{mr}-V_{th}}{R_{dson}})$$
(14)$$\begin{array}{rr} P_{turn\_on\_vf} & =\frac{1}{2}(i_{sta}+{\bar{I}}_{v}{}_{f})(V_{r}-V_{mr}+V_{th})(t_{2}-t_{3})f_{s}\\ & +\frac{1}{2}C_{stray}[{V_{r}}^{2}-{\left(V_{r}-V_{mr}+V_{th}\right)}^{2}]f_{s} \end{array}$$where ${\bar{I}}_{vf}$ is the average channel current during the *t*_2_–*t*_3_ time interval.3.During the *t*_3_–*t*_4_ time interval, the HEMT device operates in an ohmic conducting state. Then, *V_drain_* continues to decrease until it reaches a low on-voltage (*V_on_*) from (*V_mr_* − *V_th_*). Assuming that *i_sta_* and the Miller voltage *V_mr_* do not change, the *t*_3_–*t*_4_ time interval, *V_on_r_*, and the power losses in this time interval (*P_turn_on_mr_*) can be written as follows [36]:(15)$$t_{3}-t_{4}=\frac{Q_{gd}R_{g\_on}}{V_{drive\_H}-V_{mr}}$$
(16)$$V_{on\_r}=i_{sta}R_{dson}k_{dv}k_{df}k_{dd}k_{th\_R}k_{cu}$$
(17)$$\begin{array}{ll} P_{turn\_on\_mr} & =\frac{1}{2}i_{sta}(V_{mr}-V_{th}-V_{on\_r})(t_{3}-t_{4})f_{s}\\ & \;\;+\frac{1}{2}C_{stray}[{\left(V_{mr}-V_{th}-V_{on\_r}\right)}^{2}-{V_{on\_r}}^{2}]f_{s} \end{array}$$


From the above analysis, Equation (6) can be modified as follows:(18)$$P_{turn\_on}(\mathrm{measured})=P_{turn\_on\_cr}+P_{turn\_on\_vf}+P_{turn\_on\_mr}$$

We noticed that at this stage, *i_sta_* is a tested drain current instead of a real channel current, and they are actually different in the *t*_2_–*t*_3_ time interval, as shown in Figure 5a. However, *I_channel_* is the real factor that results in the power losses in this stage, and the real *I_channel_* is the combined current of *I_drain_* and the discharging current of *C_oss_*:(19)$$I_{channel}=I_{drain}+I_{Cds}+I_{Cgd}\approx I_{drain}+I_{Cds}$$

Thus, Equation (18) can be finally modified as follows [12]:(20)$$P_{turn\_on\_act}=P_{turn\_on\_mea}+P_{turn\_on}{}_{\_dis}$$
where
(21)$$P_{turn\_on}{}_{\_dis}=\frac{1}{2}C_{oss}V_{ds\_\mathrm{off}}{}^{2}f_{s}$$

Thus,
(22)$$\begin{array}{ll} P_{turn\_on}{}_{\_act} & =P_{turn\_on\_cr}+P_{turn\_on\_vf}\\ & \;\;\;\;+P_{turn\_on\_mr}+\frac{1}{2}C_{oss}V_{dsf}{}^{2}f_{s} \end{array}$$

### 6.4. Stage 4 (S4)—Turn-off Transition

During the *t*_7_–*t*_11_ time intervals, the device is in a turn-off transition. In the *t*_7_–*t*_8_ time intervals, the drain voltage increases, while *I_drain_* stays almost constant; in the *t*_8_–*t*_9_ time intervals, the drain voltage continuously increases, while *I_drain_* slightly decreases. In the *t*_9_–*t*_10_ time intervals, *I_drain_* decreases, while the drain voltage stays almost constant. Finally, *I_drain_* decreases to zero, and the drain voltage becomes resonant in the *t*_10_–*t*_11_ time intervals. These crossovers of *V_drain_* and *I_drain_* will cause power losses during the turn-off transition (*P_turn_off_*) as follows:(23)$$P_{turn\_off}=\int_{t_{7}-t_{10}}V_{ds}(t)I_{drain}(t)f_{s}dt$$


4.In the *t*_7_–*t*_8_ time interval, the observations are very similar to those in the *t*_3_–*t*_4_ time interval. The HEMT device moves into a linear region from an ohmic conducting state. *V_drain_* increases to a boundary voltage of $V_{mf}-V_{th}$. Assuming that the peak current is unchanged, and $V_{mf}=V_{mr}$, the *t*_7_–*t*_8_ time interval, *V_on_f_*, and the power losses in this time interval (*P_turn_on_mf_*) can be written as follows:(24)$$t_{7}-t_{8}=\frac{Q_{gd}R_{g\_off}}{V_{mf}-V_{drive\_L}}$$
(25)$$V_{on\_f}=I_{pk}R_{dson}k_{dv}k_{df}k_{dd}k_{th\_R}k_{cu}$$
(26)$$P_{turn\_off\_mf}=\frac{1}{2}i_{pk}(t_{7}-t_{8})(V_{mf}-V_{th}-V_{on\_f})f_{s}$$5.In the *t*_8_–*t*_9_ time interval, the observations are very similar to those in the *t*_2_–*t*_3_ time intervals. *V_drain_* continues to increase more quickly towards the off-state *V_ds_off_*, while *I_drain_* decreases slightly to *i_r_*. This current drop is caused by a charging shunt to other peripheral devices [33], and the current path through the device is illustrated in Figure 5d. Assuming that the Miller voltage (*V_mf_*) remains unchanged and the current-dependent charging time of *C_oss_* can no longer be ignored, we have the following equation:(27)$$\begin{array}{l} t_{8}-t_{9}\approx \frac{Q_{gd}R_{g\_off}+C_{stray}(V_{ds}-V_{mf}+V_{th})/(2g_{m})}{V_{mf}-V_{drive\_L}}\\ \;\;\;\;\;\;\;\;\;\;\;\;\;\;\;+\frac{C_{oss}(V_{ds}-V_{mf}+V_{th})}{i_{pk}} \end{array}$$
(28)$$i_{r}=i_{pk}-C_{stray}\frac{dV_{ds}}{dt}=i_{pk}-C_{stray}\frac{V_{ds}-V_{mf}+V_{th}}{t_{8}-t_{9}}$$
(29)$$P_{turn\_off\_vr}=\frac{i_{pk}+i_{r}}{2}(V_{ds}+V_{mr}-V_{th})(t_{8}-t_{9})f_{s}$$6.In the *t*_9_–*t*_10_ time interval, the observations are similar to those in the *t*_1_–*t*_2_ time interval. *I_drain_* decreases from *i_r_* to a low value because the current begins to divert from the HEMT device to D_1_. In this time interval, the drain voltage is in a state of resonance, while *V_gs_* decreases to (*V_mr_* − *V_th_*), and the device channel current reaches zero at *t*_10_ [20]. Then, the *t*_9_–*t*_10_ time interval and the power losses at this time interval (*P_turn_off_cf_*) can be written as follows:(30)$$t_{9}-t_{10}=\frac{(C_{gs}R_{g\_off}+L_{s}g_{m})i_{r}}{[0.5(V_{mf}+V_{th})-V_{drive\_L}]g_{m}}$$
(31)$$P_{turn\_off\_cf}=\frac{1}{2}i_{r}V_{ds\_off}(t_{9}-t_{10})f_{s}+\frac{L_{stray}i_{r}{}^{2}}{2}$$7.During the *t*_10_–*t*_11_ time interval, the device is turned off, but *V_drain_* ringing occurs due to the resonance between *C_oss_* and *L_stray_*. These fluctuations of the drain voltage will lead to a slight power loss, which depends on the ringing peak voltage (*V_ds_pk_*). Assuming that the reverse recovery of D_1_ is zero, we have the following equation:(32)$$\frac{L_{stray}i_{r}{}^{2}}{2}=\frac{C_{oss}\Delta V^{2}}{2}\to \Delta V=\sqrt{\frac{L_{stray}i_{r}{}^{2}}{C_{oss}}}\to V_{ds\_pk}=V_{ds\_off}+\Delta V$$
(33)$$P_{turn\_off\_vx}\approx \frac{1}{2}C_{oss}(V_{ds\_pk}{}^{2}-V_{ds\_off}{}^{2})f_{s}$$


From the above analysis, Equation (23) can be modified as follows:(34)$$P_{turn\_off}(\mathrm{measured})=P_{turn\_off\_mf}+P_{turn\_off\_vr}+P_{turn\_off\_cf}+P_{turn\_off\_vx}$$

Instead of real channel currents, they are actually different in the *t*_8_–*t*_9_ time interval, as shown in Figure 5c. However, *I_channel_* is the real factor that results in the power losses in this stage, and the real *I_channel_* is the diverted current of *I_drain_* and the charging current of *C_oss_*
(35)$$I_{channel}=I_{drain}-I_{Cds}-I_{Cgd}\approx I_{drain}-I_{Cds}$$

Therefore, Equation (34) can be finally modified as follows [12]:(36)$$P_{turn\_off}(\mathrm{actual})=P_{turn\_off}(\mathrm{measured})-P_{turn\_off}(\mathrm{charge})$$
where
(37)$$P_{turn\_off\_char}=\frac{1}{2}C_{oss}V_{ds\_off}{}^{2}f_{s}$$

Thus,
(38)$$\begin{array}{l} P_{turn\_off\_act}=P_{turn\_off\_mf}+P_{turn\_off\_vr}+P_{turn\_off\_cf}\\ \;\;\;\;\;\;\;\;\;\;\;\;\;\;\;\;\;\;\;\;\;+P_{turn\_off\_vx}-\frac{1}{2}C_{oss}V_{ds\_off}{}^{2}f_{s} \end{array}$$

Finally, the total power loss (*P_total_*) should be described based on the sum of Equations (3)–(38):(39)$$P_{total}=P_{off}+P_{con}+P_{turn\_on}{}_{\_act}+P_{turn\_off}{}_{\_act}$$

In particular, the effects of *I_channel_* and *I_drain_* on *P_total_* can finally cancel out for a hard switch. However, in a soft switch application, such as a zero-voltage switch (ZVS), $P_{turn\_on}{}_{\_dis}$ is zero; hence,$P_{turn\_off}{}_{\_char}$ can no longer cancel out. This correction becomes very meaningful to the universality of the dynamic power loss model for eGaN HEMTs.

As can be seen, *P_total_* in Equation (39) is very different to that in Equation (1). Equation (39) has no power loss of reverse recovery, but it takes the trapping effect-induced dynamic *R_dson_* and the impacts of the *I_drain_* and the real *I_channel_* into account.

## 7. Model Verification via Experiments

To verify our dynamic power loss model, we adopt a floating buck–boost power converter with a light-emitting diodes (LEDs) operating in DCM and CCM. To maintain the operation mode and the output current (*I_o_*) in an open-loop control system, some key parameters are adjusted (such as L_1_) or tested (such as the output voltage *V_o_*, the peak operating current *I_pk_*, and the output power *P_o_*) in the circuit, as shown in Figure 9, for an input voltage (*V_Bulk_*) of 400 V, a duty cycle of 10%, and various *f_s_* and *I_o_* values.

The power losses are then tested using a power analyzer (PW6001-03 from HIOKI Inc., Ueda, Nagano Prefecture, Japan). Figure 10a–c reveal that the analytical results of the total dynamic power losses generated via the proposed model are in good agreement with experimental results in both CCM and DCM, even for various *I_o_*, *f_s_* and *V_Bulk_* values. The experimental results are slightly different from the analytical results, which may be because of the measurement accuracy of the power meter reduced at a high *f_s_*.

Figure 11a shows the relationship between the total dynamic power losses and *I_o_* in CCM and DCM. In the case of a small *I_o_*, the switching loss is dominant, while in the case of a large *I_o_*, the conduction loss is dominant. In addition, the dynamic power loss increases faster with the increase in *I_o_* in DCM than in CCM, indicating that DCM is not suitable for high current conditions.

Figure 11b shows the switching loss during the turn-on and turn-off transitions in CCM and DCM. The results reveal that the switching loss is lower in DCM than in CCM during the turn-on transition, but larger during the turn-off transition when *I_o_* is larger than 1.25 A. This finding means that DCM is more suitable for a relatively small *I_o_*. In this case, according to Figure 10b and Figure 11a, a 1.25-ampere *I_o_* is a moderate output current that is acceptable.

It also can be seen that all calculated results are a little bit higher than the experimental results, especially in a higher than 2-ampere *I_o_* condition and CCM mode. According to the analysis, the calculation model is not accurate enough to evaluate the self-heating effect of the device. Of course, in a high-frequency circuit, the peak current and the operating temperature of the device should be controlled by designing a suitable heat sink. In this case, when the *I_o_* is 2 A, the device peak current is as high as 10 A, which is higher than the rated device continuous operating current of 7.5 A. Generally, our design should ensure that the operating current of the device does not exceed the rated 7.5 A, a certain margin should be designed, and the operating temperature of the device should not exceed 120 °C. Although the high operating temperature may not have any effect on the GaN device, it will have a bad effect on other surrounding devices.

To restrain the peak current and obtain a high operating efficiency, the QRM is, thus, proposed. The reason for this proposal is that QRM works at the DCM boundary, where *V_drain_* will decrease to a minimum value at the beginning of the turn-on transition, while *I_drain_* decreases to zero. In addition, the peak on-state current in DCM is usually larger than that in CCM, and the current will be at a controllable high level, meaning that the turn-off time is fast in QRM. Therefore, QRM is more suitable for the achievement of a lossless switch and even for the reduction in the turn-off transition time.

## 8. Conclusions

An improved 12-time-interval piecewise dynamic power loss model for eGaN HEMTs is developed by specially quantifying the effects of the increase in the dynamic *R_dson_*, the impact of *I_drain_* on the turn-on and turn-off times, and the real *I_channel_*; good agreement with some experimental results is proven.

In this work, three methods or techniques are proposed, which are DDI method, the double-mode test technique and qualitative method. Then, the dynamic *R_dson_* is obtained by our new extraction circuit at a high operating *fs* of up to 1 MHz and high drain voltage of up to 600 V, the drain current is found to significantly affect the turn-on and turn-off times via a switching circuit operating in DCM and CCM, and the real channel current is accurately calculated to distinguish it from the measured drain current. All of these parameters are included in the power loss model.

Moreover, the QRM has a low *V_ds_off_*, zero turn-on current, and relatively large peak current. Therefore, on the basis of the model, we propose the QRM to obtain a high efficiency and decrease the turn-off switching time required for the application of eGaN HEMTs.

## Figures and Tables

**Figure 1 micromachines-14-01633-f001:**
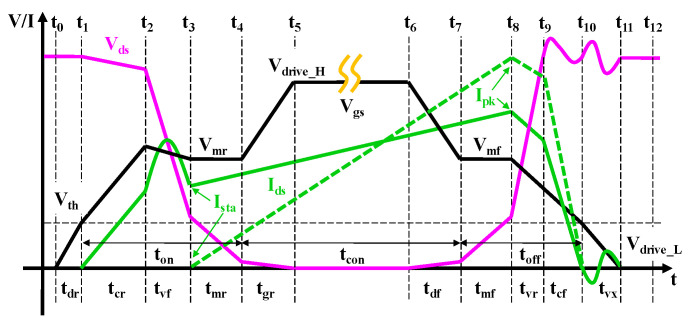
Piecewise timing diagram of the power switching devices.

**Figure 2 micromachines-14-01633-f002:**
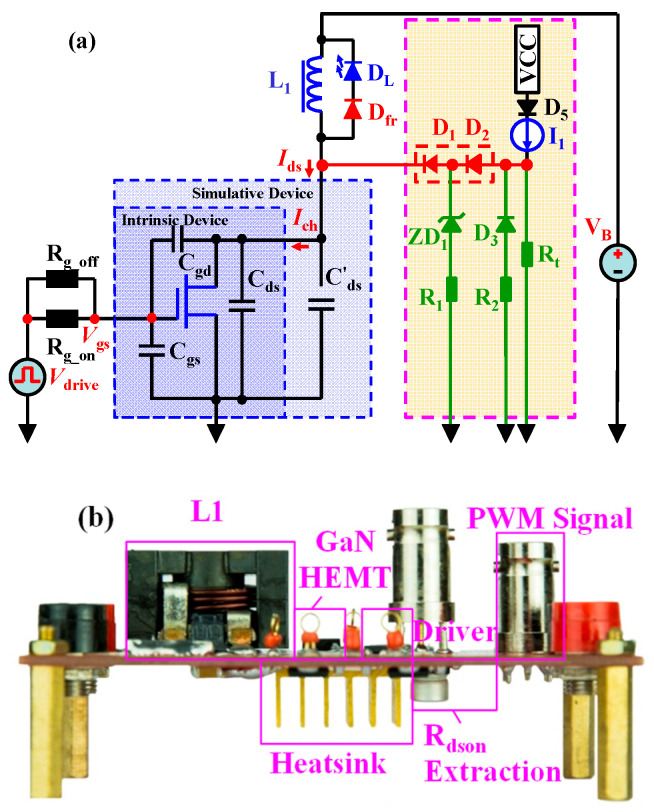
Lumped equivalent switching circuit with a floating buck–boost topology (**a**), and a photograph of the assembled printed circuit board (**b**).

**Figure 3 micromachines-14-01633-f003:**
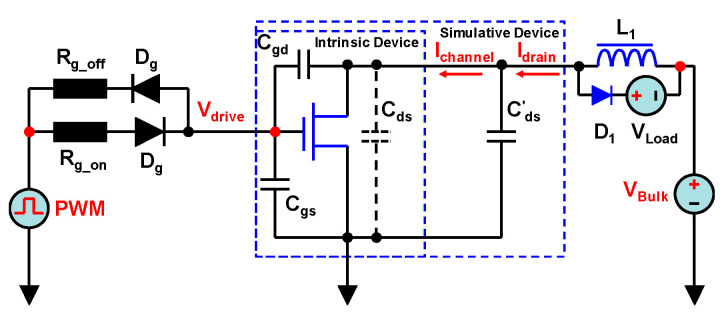
The lumped simulation circuit using an extra parasitic capacitor.

**Figure 4 micromachines-14-01633-f004:**
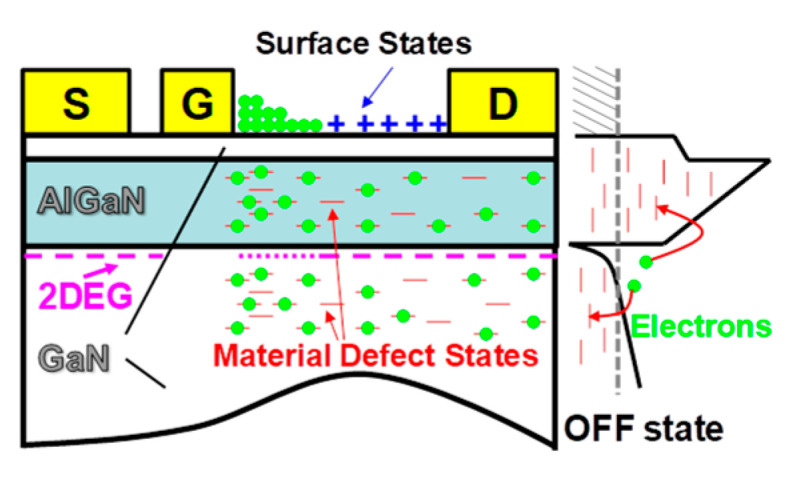
Mechanism of the dynamic *R_dson_*.

**Figure 5 micromachines-14-01633-f005:**
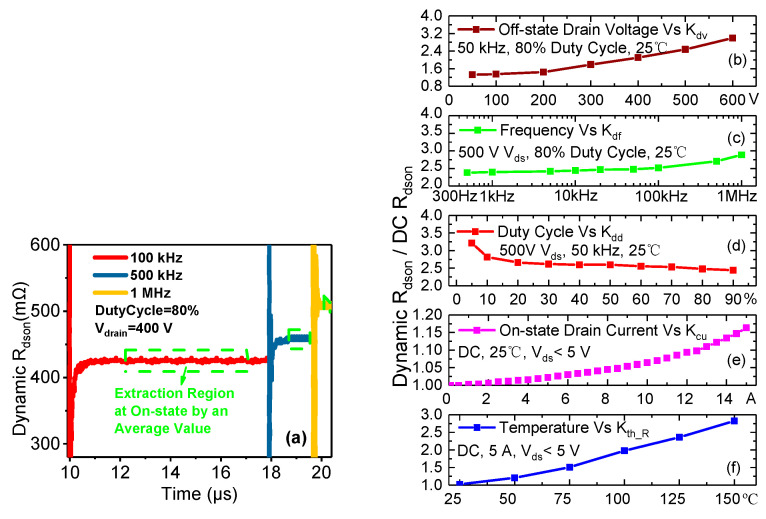
Dynamic *R_dson_* extraction waveforms at various *f_s_* (**a**) and the dynamic *R_dson_* normalized by *R_dson_DC_* for various *V_ds_off_* (**b**), *fs* (**c**), duty cycles (**d**), *I_drain_* (**e**), and temperatures (**f**).

**Figure 6 micromachines-14-01633-f006:**
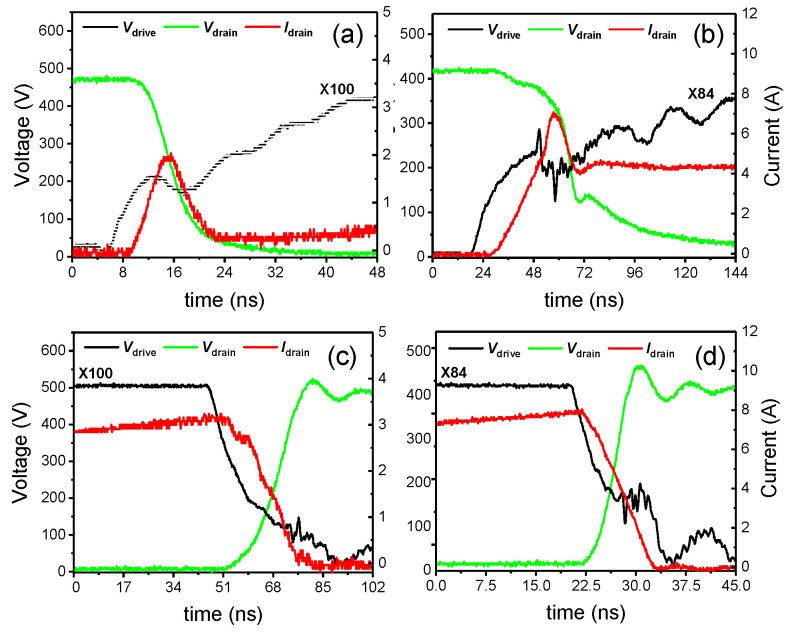
Experimental waveforms of the HEMT device during the turn-on transitions in 400-volt DCM with a *V_Load_* of 80 V (**a**) and 400-volt CCM with a *V_Load_* of 20 V (**b**), as well as during turn-off transitions in 400-volt DCM with a *V_Load_* of 80 V (**c**) and 400-volt CCM with a *V_Load_* of 20 V (**d**).

**Figure 7 micromachines-14-01633-f007:**
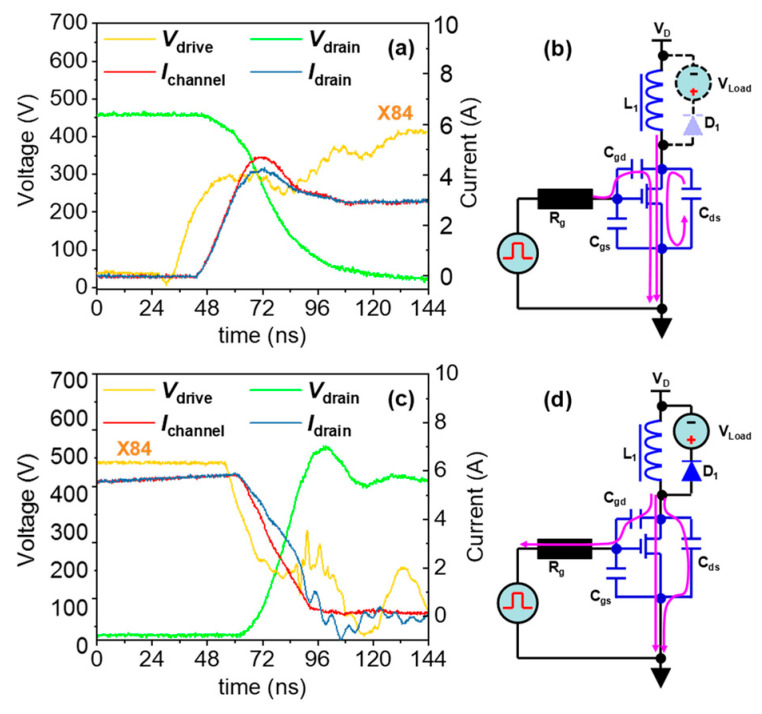
Experimental results during the turn-on transition in 500-volt CCM (**a**) and a schematic diagram of the corresponding current path (**b**), and the experimental results during the turn-off transition in 500-volt CCM (**c**) and a schematic diagram of corresponding current path (**d**).

**Figure 8 micromachines-14-01633-f008:**
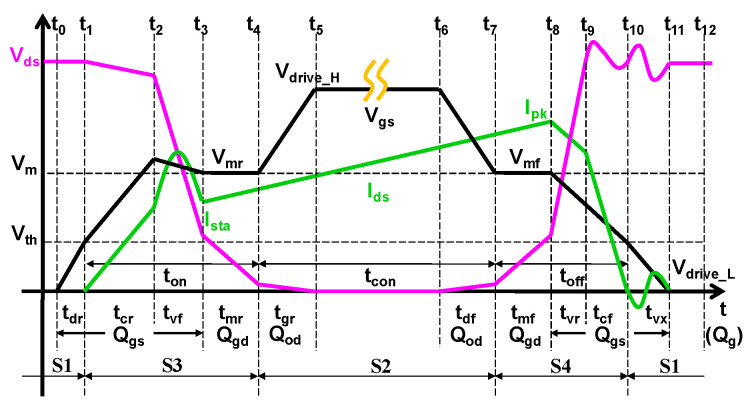
Timing diagram of the GaN HEMT devices.

**Figure 9 micromachines-14-01633-f009:**
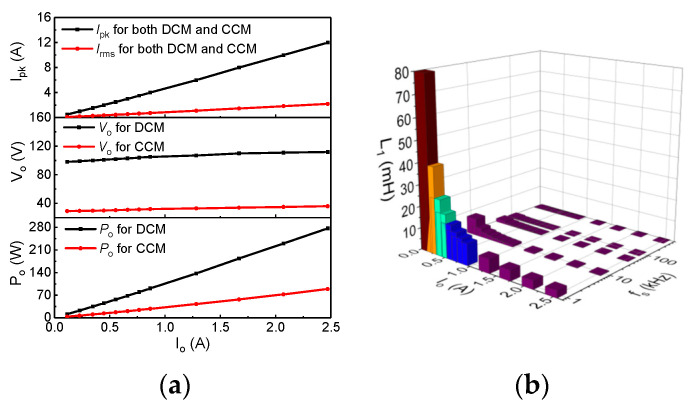
The relationship between *Io* and *V_o_* and *P_o_*, and *I_pk_* (**a**), and the relationship between *I_o_* and inductance of L_1_ for various *f_s_* (**b**) in an open-loop-controlled floating buck–boost power converter.

**Figure 10 micromachines-14-01633-f010:**
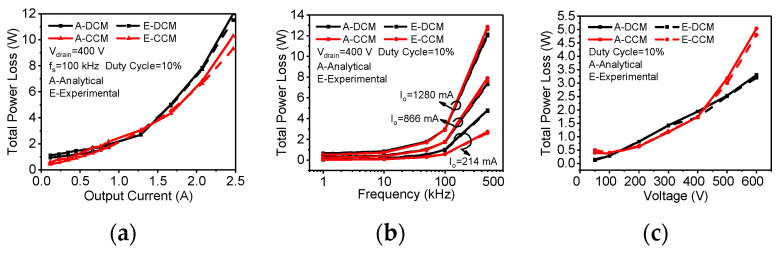
Comparison between the total dynamic power losses from the analytical and experimental results in both CCM and DCM and for various *I_o_* (**a**), *f_s_* (**b**) and *V_Bulk_* (**c**).

**Figure 11 micromachines-14-01633-f011:**
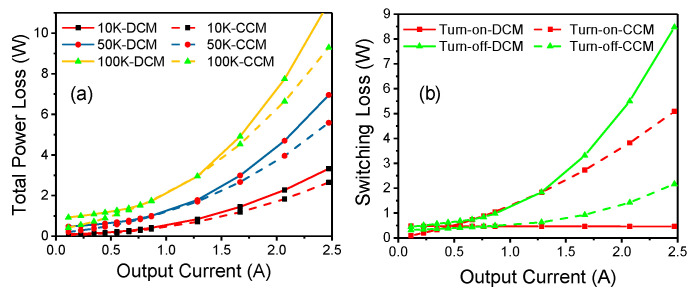
Experimental total dynamic power losses (**a**) and switching losses (**b**) as a function of the output current in DCM and CCM.

## Data Availability

Data are available by request to the corresponding author.

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
