# Peer review of "An Analytical Model of Dynamic Power Losses in eGaN HEMT Power Devices"

_micromachines, 2023, doi:10.3390/mi14081633_

Round 1

Reviewer 1 Report

Overall, the manuscript presents an analytical model of dynamic power losses for enhancement-mode AlGaN/GaN high-electron-mobility transistor power devices (eGaN HEMTs). The work is interesting and potentially valuable for the field, but there are some areas that need improvement and clarification. Below are specific comments to enhance the manuscript:

Introduction: The introduction should provide a more comprehensive literature review to contextualize the importance of the proposed dynamic power loss model for eGaN HEMTs. Include relevant studies and explain how your model addresses existing challenges or gaps in the literature.

Methods and Experiments: a. Provide more details on the "double-diode isolation (DDI)" method used for the accurate extraction of dynamic on-resistance (Rdson). Include circuit diagrams and equations to help readers understand the methodology better.
b. Similarly, explain the "dual-pulse-current-mode test (DPCT)" in detail, particularly the discontinuous conduction mode (DCM) and continuous conduction mode (CCM) aspects. This will help readers replicate the experiments if necessary.
c. The "external shunt capacitance (ESC)" method for investigating the discrepancy between drain current and real channel current should be elaborated upon. Clarify how this method overcomes challenges related to device parasitic capacitance and why it is suitable for this analysis.

Results and Discussion: a. Provide a clearer explanation of the discrepancies observed between the dynamic model and experimental results. Are there specific areas where the model excels or falls short compared to experimental data?
b. Discuss the practical implications and limitations of your proposed dynamic power loss model. How can it be applied in real-world scenarios, and are there any constraints in its application?
c. Address the impact of various operating conditions (e.g., frequency, voltage) on the model's accuracy. Do certain conditions lead to higher discrepancies between the model and experimental results?

minor

Author Response

Response to Reviewer 1 Comments

Point 1: Introduction: The introduction should provide a more comprehensive literature review to contextualize the importance of the proposed dynamic power loss model for eGaN HEMTs. Include relevant studies and explain how your model addresses existing challenges or gaps in the literature.

Response 1: We have revised the literature review in the introduction section, expanded the description of the advantages and disadvantages of the cited literatures, and clearly pointed out the importance of our model and the problems to be solved by this model.

“Recently, some state-of-the-art dynamic power loss models for eGaN HEMTs have been proposed. Wang et al. developed two analytical loss models based on detailed parasitic parameters for high-voltage and low-voltage GaN eHEMTs [4], [5]. In these models, the gate charge (Qg) and output charge (Qoss) instead of the voltage-dependent capacitance were used to improve the nonlinear characteristics. However, the loss caused by output capacitance is not discussed separately in a hard switch and soft switch. Shen et al. fully accounted for the effects of parasitic parameters and transconductance [6]. Hou et al. and Guacci et al. investigated the losses caused by the output capacitance in a hard switch and soft switch using simulation methods rather than experimental methods, respectively [7], [8]. However, these models did take into account the impact on device loss from some aspects instead of fully evaluated, the results accuracy is affected. For example, the problem of increased losses caused by dynamic on-resistance (Rdson) is not discussed.

Chen et al. presented a complete analytical loss model for low-voltage eGaN HEMTs, which a piecewise model was employed [9]. Piecewise models are also usually used to evaluate the dynamic power losses for Si- and SiC-based metal-oxide-semiconductor field-effect transistors (MOSFETs) [10]-[15]. These models are carefully considered and allow accurate evaluation of device power losses. However, these models did not take into account the exclusive dynamic physical characteristics of eGaN HEMTs, such as a strong defect trapping-effect, increased dynamic on-resistance (Rdson), and no reverse body diode. fail to include sufficient consideration of the parasitic elements, or are merely focusing on Si-based MOSFETs, not on GaN HEMTs. In addition, the effect of current operation mode on device transition time and loss is not considered in all known device loss models. Therefore, these analytical models need are necessary to be modified for eGaN HEMTs to make a more comprehensive and accurate model. In addition, the effect of current operation mode on device transition time and loss is not considered in all known device loss models.

Point 2: Methods and Experiments:

  1. Provide more details on the "double-diode isolation (DDI)" method used for the accurate extraction of dynamic on-resistance (Rdson). Include circuit diagrams and equations to help readers understand the methodology better.
    b. Similarly, explain the "dual-pulse-current-mode test (DPCT)" in detail, particularly the discontinuous conduction mode (DCM) and continuous conduction mode (CCM) aspects. This will help readers replicate the experiments if necessary.
    c. The "external shunt capacitance (ESC)" method for investigating the discrepancy between drain current and real channel current should be elaborated upon. Clarify how this method overcomes challenges related to device parasitic capacitance and why it is suitable for this analysis.

Response 2:

  1. “The part of Fig. 2(b) marked with the light-yellow area shows our novel dynamic Rdson extraction circuit based on a double-diode isolation (DDI) method, the details on how to configure, test and calculate can be obtained from our Ref. [17]. “

b1. “In the piecewise model, the operating sequence of the device is shown in figure 1. In particular, the device is usually connected in series with to a choke or a transformer in power switches, thus the value of Ista indicate the current mode of device. (Ista=0) denotes the device operate in discontinuous conduction mode (DCM), while (Ista>0) denotes the device operate in continuous conduction mode (CCM). Whether the device operates in CCM or DCM depends on the choke in series. As we all know, in high-frequency circuit, chokes follow the volt-second balance principle, that is, the starting current of the choke in each cycle must be equal to the end current. During the ton time when the device is on, the choke current rises under an applied forward voltage V1, the rise slope is V1/L, and L is the inductance of the choke; while during the toff time when the device is off, the choke current falls under an applied reverse voltage V2, the fall slope is V2/L. Therefore, the peak value of the choke current in each cycle is V1·ton /L, and the end current value will be (V1·ton - V2·toff) /L. Then, according to the known values of V1 and V2, we can design the required L to make the device operate in CCM or DCM. “

b2. In general, the electrical performance of GaN devices is characterized by single pulse or double-pulse mode. The typical waveforms of the “double-pulse” test are shown as below. The first step, represented by turn-on pulse number 1, is the initial adjusted pulse width. This establishes current in the inductor. This pulse is adjusted to get to the desired test current (Id). The second step (2) is to turn-off the first pulse, which creates current in the free-wheeling diode. The turnoff period is short to keep the load current as close as possible to a constant value through the inductor, and the Id on the low side MOSFET that goes to zero in step 2; however, the current is flowing through the inductor and the high side diode. The third step (3) is represented by the second turn-on pulse. The pulse width is shorter than the first pulse so that the device is not overheated. The second pulse needs to be long enough for the measurements to be taken. Turn-off and turn-on timing measurements are then captured at the turn-off of the first pulse and the turn-on of the second pulse. This “double-pulse” technique only sends 2 pulses to the device, which is not periodically sustained, and the current in step (3) is always higher than 0 A.

In order to fully obtain the characteristics of the periodic operation of the device in the high frequency circuit, we make the device continuously working periodically and stably in the CCM or DCM state by controlling the L value and the reverse voltage V2 during the toff time. This detailed method has shown in the Response 2-b1. With the “double-pulse-current-mode” technique, we are able to focus on the impact of the starting current and peak current on the transition time of the device, which is not easy to do with the conventional “double-pulse” technique.

  1. The "external shunt capacitance (ESC)" method has been elaborated on section 1-C and Fig. 3.

“Since we cannot perform the measurements inside the HEMT device directly, we propose an evaluation method by employing an extended parallel capacitor Cds', as shown in Fig. 3, which we call it "external shunt capacitance (ESC)" method. “

Point 3: Results and Discussion:

  1. Provide a clearer explanation of the discrepancies observed between the dynamic model and experimental results. Are there specific areas where the model excels or falls short compared to experimental data?
    b. Discuss the practical implications and limitations of your proposed dynamic power loss model. How can it be applied in real-world scenarios, and are there any constraints in its application?
    c. Address the impact of various operating conditions (e.g., frequency, voltage) on the model's accuracy. Do certain conditions lead to higher discrepancies between the model and experimental results?

Response 3:

a/b.” The power losses are then tested by a power analyzer (PW6001-03 from HIOKI Inc.). Figs. 10(a)-(c) reveal that the analytical results of the total dynamic power losses by the proposed model are in good agreement with experimental results both in CCM and DCMs, even for various Io, fs and VBulk values. The experimental results are slightly different from the analytical results, which may be because of the measurement accuracy of the power meter reduced at a high fs.

Fig. 11(a) shows the relationship between the total dynamic power losses and Io in CCM and DCMs. In the case of a small Io, the switching loss is dominant, while in the case of a large Io, the conduction loss is dominant. In addition, the dynamic power loss increases faster with the increase in Io in DCM than in CCM, indicating that DCM is not suitable for high current conditions.

Fig. 11(b) shows the switching loss during the turn-on and turn-off transitions the total dynamic power losses in CCM and DCM. The results reveal that the switching loss is lower in DCM than in CCM during the turn-on transition but larger during the turn-off transition when Io is larger than 1.25 A. This means that DCM is more suitable for a relatively small Io. In this case, from Fig. 11 (a) and Fig. 10(b), 1.25 A Io is a moderate output current that is acceptable.

Fig. 11(c) shows the relationship between the total dynamic power losses and VBulk. It can be seen that the higher the VBulk, the higher the switching loss.

It also can be seen that all calculated results are a little bit higher than the experimental results, especially in higher than 2 A Io condition and in CCM mode. According to the analysis, the calculation model is not accurate enough to evaluate the self-heating effect of the device. Of course, in a high-frequency circuit, the peak current and the operating temperature of the device should be controlled by designing a suitable heat sink. In this case, when the Io is 2 A, the device peak current is up to 10 A, which is higher than the rated device continuous operating current of 7.5 A. Generally, our design should ensure that the operating current of the device does not exceed the rated 7.5 A, and a certain margin should be designed, and the operating temperature of the device should not exceed 120℃. Although the high operating temperature may not have any effect on the GaN device, it will also have a bad effect on other surrounding devices.”

  1. Due to the very complex operating conditions of the device in the high-frequency circuit, It’s very hard for us to accurately address the influence of a certain condition on the total dynamic power losses. Our test only selected some typical specific points for experimental verification, and then inferred the difference between the calculated model and the experimental data through curve fitting. From Figs. 10(a)-(c), we can see that the analytical results of the total dynamic power losses by the proposed model are in good agreement with experimental results both in CCM and DCMs, even for various Io, fs and VBulk values.

Point 4: Comments on the Quality of English Language:minor

Response 4: We have modified some grammar problem and have improved some English expression to make it more readable.

Reviewer 2 Report

The following phrase of the Introduction: "the effect of current operation mode on transition time is revealed for the first time and thus the loss of the device is affected;" is a rather strong statement, since the trap-related effects on GaN are well-known since decades. Please eliminate or rephrase.

There's a sufficient list of references and literature study related to the modeling, but not enough detailed with respect to the experimental setups, settings, etc. which is also very important in this kind of evaluation, see for example Alemanno, A. et al. A Reconfigurable Setup for the On-Wafer Characterization of the Dynamic RON of 600 V GaN Switches at Variable Operating Regimes. Electronics 2023, 12, 1063. I'd suggest to add a paragraph for the literature study for the experimental benches and techniques. 

The list of parameters in Section 2 could be moved to an Appendix. The current axis is not indicated in Fig. 1 (only the voltage axis is indicated). There is a wrong referencing to Fig. 1(b) a line 130. Also Fig. 5(b) at line 223 seems to be a wrong referencing.

In Sec. 4, the authors mention "a new double-mode test technique" but it is not clear what is new. Could the authors also better explain and reference the "dual-pulse-current-mode test" and the correspondence with the "double-pulse" technique found in literature?

It would be important to include the experimental validation of the dynamic losses at different duty cycles and different drain voltages.

The language is understandable although it could be improved with a proofread by an English mother tongue.

Author Response

Response to Reviewer 2 Comments

Point 1: The following phrase of the Introduction: "the effect of current operation mode on transition time is revealed for the first time and thus the loss of the device is affected;" is a rather strong statement, since the trap-related effects on GaN are well-known since decades. Please eliminate or rephrase.

Response 1: We revised the statement in accordance with the reviewer’s suggestion.

“To improve the dynamic power loss model of eGaN HEMTs, we propose three experimental methods according to the practical application of devices in high-fs circuits, such as a double-diode isolation (DDI) method, a dual-pulse-current-mode test (DPCT) method, and an external shunt capacitance (ESC) method. Then, the dynamic Rdson is accurately extracted in a high operating frequency (fs) up to 1 MHz and a high drain voltage up to 600 V; the effect of current operation mode on transition time is revealed for the first time and thus the loss of the device is affected the effect of current operation mode on the device loss is discussed from the shape of operating waveforms in the circuit; and the real channel current (Ich) is qualitative modified compared to the drain current (Idrain) we can directly test in device drain side. Afterward the dynamic power loss of eGaN HEMTs is carefully described by a modified 12-segment piecewise model. Finally, we propose a quasi-resonant mode (QRM) with a low off-state drain voltage (Vds_off), zero turn-on current, and a relatively large on-state peak current for a lossless design and fast transit speed in power switches. “

Point 2: There's a sufficient list of references and literature study related to the modeling, but not enough detailed with respect to the experimental setups, settings, etc. which is also very important in this kind of evaluation, see for example Alemanno, A. et al. A Reconfigurable Setup for the On-Wafer Characterization of the Dynamic RON of 600 V GaN Switches at Variable Operating Regimes. Electronics 2023, 12, 1063. I'd suggest to add a paragraph for the literature study for the experimental benches and techniques.

Response 2:

  1. “The part of Fig. 2(b) marked with the light-yellow area shows our novel dynamic Rdson extraction circuit based on a double-diode isolation (DDI) method, the details on how to configure, test and calculate can be obtained from our Ref. [17]. “
  2. “In the piecewise model, the operating sequence of the device is shown in figure 1. In particular, the device is usually connected in series with to a choke or a transformer in power switches, thus the value of Ista indicate the current mode of device. (Ista=0) denotes the device operate in discontinuous conduction mode (DCM), while (Ista>0) denotes the device operate in continuous conduction mode (CCM). Whether the device operates in CCM or DCM depends on the choke in series. As we all know, in high-frequency circuit, chokes follow the volt-second balance principle, that is, the starting current of the choke in each cycle must be equal to the end current. During the ton time when the device is on, the choke current rises under an applied forward voltage V1, the rise slope is V1/L, and L is the inductance of the choke; while during the toff time when the device is off, the choke current falls under an applied reverse voltage V2, the fall slope is V2/L. Therefore, the peak value of the choke current in each cycle is V1·ton /L, and the end current value will be (V1·ton - V2·toff) /L. Then, according to the known values of V1 and V2, we can design the required L to make the device operate in CCM or DCM. “
  3. The circuit schematic and the prototype are shown in section 1-B and Fig. 2, and all the key parameters in the schematic are also presented in this chapter. Then, in the subsequent experiments, we have also listed the key parameter settings, such as drain voltage, operating frequency, duty cycle, etc. For examples:
  • “To ensure that the switching circuit operates in open-loop CCM and DCM, VBulk is set to 400 V, and the VLoad is set to 80 V in DCM and 20 V in CCM, so the Vds_off values of the devices in the two modes are different. “
  • “According to the qualitative method in Fig. 2, we can study the discrepancy between Idrain and Ichannel. Fig. 7(a) shows the tested Idrain, Ichannel, Vdrain and Vdrive values of the AlGaN/GaN HEMT in the turn-on transition for a Vds_off of 500V, an fs of 100 kHz, and a duty cycle of 16.5%.“
  • “ 7(c) shows the tested Idrain, Ichannel, Vdrain and Vdrive values of the AlGaN/GaN HEMT during the turn-off transition for a Vds_off of 500 V, an fs of 100 kHz, and a duty cycle of 16.5%.“
  • “To verify our dynamic power loss model, we adopt a floating buck-boost power con-verter with a light-emitting diodes (LEDs) operating in DCM and CCMs. To maintain the operation mode and the output current (Io) in an open-loop control system, some key pa-rameters are adjusted (such as L1) or tested (such as the output voltage Vo, peak operating current Ipk, and the output power Po) in the circuit, as shown in Fig. 8, for an input voltage (VBulk) of 400 V, a duty cycle of 10%, and various fs and Io values. “

In particular, in the validation experiments shown if Fig. 9 and Fig. 10, we have given detailed drain voltage, operating frequency, duty cycle, output current, and the value of L involved in controlling CCM or DCM.

Point 3: The list of parameters in Section 2 could be moved to an Appendix. The current axis is not indicated in Fig. 1 (only the voltage axis is indicated). There is a wrong referencing to Fig. 1(b) a line 130. Also Fig. 5(b) at line 223 seems to be a wrong referencing.

Response 3:

  1. We have moved the list of parameters from Section 2 to the Glossary section at the beginning of the manuscript.
  2. The axis in Fig. 1 have been modified.

Figure 1. Piecewise timing diagram of the power switching devices.

  1. We have modified Fig. 1(b) to Fig. 2(b) on line 130.
  2. We have modified Fig. 5(b) to Fig. 5(b) on line 223.

Point 4: In Sec. 4, the authors mention "a new double-mode test technique" but it is not clear what is new. Could the authors also better explain and reference the "dual-pulse-current-mode test" and the correspondence with the "double-pulse" technique found in literature?

Response 4:

  1. We have removed the expression of “new”.
  2. In general, the electrical performance of GaN devices is characterized by single pulse or double-pulse mode. The typical waveforms of the “double-pulse” test are shown as below. The first step, represented by turn-on pulse number 1, is the initial adjusted pulse width. This establishes current in the inductor. This pulse is adjusted to get to the desired test current (Id). The second step (2) is to turn-off the first pulse, which creates current in the free-wheeling diode. The turnoff period is short to keep the load current as close as possible to a constant value through the inductor, and the Id on the low side MOSFET that goes to zero in step 2; however, the current is flowing through the inductor and the high side diode. The third step (3) is represented by the second turn-on pulse. The pulse width is shorter than the first pulse so that the device is not overheated. The second pulse needs to be long enough for the measurements to be taken. Turn-off and turn-on timing measurements are then captured at the turn-off of the first pulse and the turn-on of the second pulse. This “double-pulse” technique only sends 2 pulses to the device, which is not periodically sustained, and the current in step (3) is always higher than 0 A.

In order to fully obtain the characteristics of the periodic operation of the device in the high frequency circuit, we make the device continuously working periodically and stably in the CCM or DCM state by controlling the L value and the reverse voltage V2 during the toff time. This detailed method has shown in the Response 2-b. With the “double-pulse-current-mode” technique, we are able to focus on the impact of the starting current and peak current on the transition time of the device, which is not easy to do with the conventional “double-pulse” technique.

Point 5: It would be important to include the experimental validation of the dynamic losses at different duty cycles and different drain voltages.

Response 5:

  1. As can be seen from Fig. 5(d), when the duty cycle is bigger than 10%, the dynamic Rdson changes very little, and usually the operation duty cycle of the device in the circuit is greater than 10%, so we did not focus on the impact of the duty cycle on the device loss.
  2. As shown in Fig.10(c), we have verified the dynamic losses at different drain voltages from 50V to 600V.

Point 6: Comments on the Quality of English Language:The language is understandable although it could be improved with a proofread by an English mother tongue.

Response 6: We have modified some grammar problem and have improved some English expression to make it more readable.

Round 2

Reviewer 2 Report

The authors have answered all questions including more details on their measurement set-up, yet they did not provide literature study in the context of the many papers concerning experimental techniques. In my opinion, the authors should enlarge the list of references concerning the experimental techniques including the double-pulse testing methods.

Author Response

Response 1: We have added 8 references to this manuscript:

  1. for Ron experimental techniques:
    [18] A. Alemanno, A. M. Anngelotti, G. P. Gibiino, A. Santarelli, E. Sangiorgi and C. Florian, "A Reconfigurable Setup for the On-Wafer Characterization of the Dynamic RON of 600 V GaN Switches at Variable Operating Regimes," Electronics, vol. 12, no. 4, 1063. Feb. 2023. doi: 10.3390/electronics12041063.
  2. for Ron experimental techniques and double-pulse testing method:
    [19] R. Li, X. K. Wu, S. Yang and K. Sheng, "Dynamic ON-State Resistance Test and Evaluation of GaN Power Devices Under Hard- and Soft-Switching Conditions by Double and Multiple Pulses," IEEE T. on Power Electr., vol. 34, no. 2, pp. 1044-1053. Feb. 2019. doi: 10.1109/TPEL.2018.2844302.
  3. for "external shunt capacitance (ESC)" method:
    [20] Guo, C. Hitchcock and T. P. Chow, "Lossless turn-off switching projection of lateral and vertical GaN power field-effect transistors," Phys. Status Solidi A, vol. 214, no. 8, 1600820, June 2017, doi:10.1002/pssa.201600820.
  4. for double-pulse testing method:
    [28] Yao and R. Ayyanar, "A Multifunctional Double Pulse Tester for Cascode GaN Devices," in IEEE Transactions on Industrial Electronics, vol. 64, no. 11, pp. 9023-9031, Nov. 2017, doi: 10.1109/TIE.2017.2694381.
  5. for double-pulse testing method:
    [29] I. Rossetto et al., "Evidence of Hot-Electron Effects During Hard Switching of AlGaN/GaN HEMTs," in IEEE Transactions on Electron Devices, vol. 64, no. 9, pp. 3734-3739, Sept. 2017, doi: 10.1109/TED.2017.2728785.
  6. for double-pulse testing method:
    [30] Double Pulse Testing Power Semiconductor Devices with a 5 or 6 Series MSO with Built-in Arbitrary Function Generator, Application Note, Tektronix, Available online: https://www.tek.com.cn/documents/application-note/double-pulse-testing-power-semiconductor-devices-with-a-5-or-6-series-mso-with-built-in-afg
  7. for CCM/DCM techniques:
    [31] S. Cabizza, G. Spiazzi and L. Corradini, "GaN-Based Isolated Resonant Converter as a Backup Power Supply in Automotive Subnets," in IEEE Transactions on Power Electronics, vol. 38, no. 6, pp. 7362-7373, June 2023, doi: 10.1109/TPEL.2023.3245561.
  8. for CCM/DCM techniques:
    [32] J. Park, Y. -S. Roh, Y. -J. Moon and C. Yoo, "A CCM/DCM Dual-Mode Synchronous Rectification Controller for a High-Efficiency Flyback Converter," in IEEE T. on Power Electronics, vol. 29, no. 2, pp. 768-774, Feb. 2014, doi: 10.1109/TPEL.2013.2256371.

Furthermore, we added the following description to the manuscript:

"In general, the electrical performance of GaN devices is characterized by single pulse or double-pulse mode. The typical “double-pulse” test is performed in three steps. The first step, represented by turn-on pulse, is the initial adjusted pulse width. This pulse is adjusted to get to the desired test current. The second step is to turn-off the first pulse. The turnoff period is short to keep the load current as close as possible to a constant value. The third step is represented by the second turn-on pulse. The pulse width is shorter than the first pulse so that the device is not overheated, but needs to be long enough for the measurements to be taken. Turn-off and turn-on timing measurements are then captured at the turn-off of the first pulse and the turn-on of the second pulse. This “double-pulse” technique only sends two pulses to the device, which is not periodically sustained, and the current in third step is always higher than 0 A [27], [28]. In order to fully obtain the characteristics of the periodic operation of the device in the high frequency circuit, we make the device continuously working periodically and stably in the CCM or DCM state by controlling the L value and the Vds_off [29], [30]. With the “double-pulse-current-mode” technique, we are able to focus on the impact of the starting current and peak current on the transition time of the device, which is not easy to do with the conventional “double-pulse” technique. "

"Therefore, the channel current (Ichannel) and Idrain can be measured directly and separately by an oscilloscope and current probes. This method is different from the traditional simulation method [], and the discrepancy between Ichannel and Idrain can be visually observed."
